# B Cells and Antibodies as Targets of Therapeutic Intervention in Neuromyelitis Optica Spectrum Disorders

**DOI:** 10.3390/ph14010037

**Published:** 2021-01-06

**Authors:** Jan Traub, Leila Husseini, Martin S. Weber

**Affiliations:** 1Department of Neurology, University Medical Center, 37075 Göttingen, Germany; traub_j1@ukw.de (J.T.); leila.husseini@med.uni-goettingen.de (L.H.); 2Department of Cardiology, University Medical Center, 97080 Würzburg, Germany; 3Institute of Neuropathology, University Medical Center, 37075 Göttingen, Germany

**Keywords:** neuromyelitis optica spectrum disorders, B cells, antibodies, eculizumab, ravulizumab, inebilizumab, tocilizumab, satralizumab, ublituximab

## Abstract

The first description of neuromyelitis optica by Eugène Devic and Fernand Gault dates back to the 19th century, but only the discovery of aquaporin-4 autoantibodies in a major subset of affected patients in 2004 led to a fundamentally revised disease concept: Neuromyelits optica spectrum disorders (NMOSD) are now considered autoantibody-mediated autoimmune diseases, bringing the pivotal pathogenetic role of B cells and plasma cells into focus. Not long ago, there was no approved medication for this deleterious disease and off-label therapies were the only treatment options for affected patients. Within the last years, there has been a tremendous development of novel therapies with diverse treatment strategies: immunosuppression, B cell depletion, complement factor antagonism and interleukin-6 receptor blockage were shown to be effective and promising therapeutic interventions. This has led to the long-expected official approval of eculizumab in 2019 and inebilizumab in 2020. In this article, we review current pathogenetic concepts in NMOSD with a focus on the role of B cells and autoantibodies as major contributors to the propagation of these diseases. Lastly, by highlighting promising experimental and future treatment options, we aim to round up the current state of knowledge on the therapeutic arsenal in NMOSD.

## 1. Introduction

Neuromyelitis optica spectrum disorders (NMOSD) are a heterogeneous group of monophasic or recurrent autoinflammatory diseases of the central nervous system (CNS). Characteristically, symptoms like severe loss of vision, weakness or paralysis of extremities, loss of sensation, bowel and bladder dysfunction, area postrema clinical syndrome or respiratory failure are the result of rapidly sequential or bilateral optic neuritis, longitudinally extensive myelitis and inflammatory brain lesions affecting the circumventricular organs [1]. The first description of a patient suffering from amaurosis associated with autoptic-proven myelitis by Antoine Portal dates back to the year 1804, while the systematic description of ‘myelitis optica’ by Devic and Gault appeared by the end of the 19th century [2]. For many years, NMOSD were considered prognostically unfavorable variants of multiple sclerosis. When specific pathogenic aquaporin-4 (AQP4) autoantibodies were detected in a major subset of patients at the beginning of the 21st century, NMOSD were finally considered to belong to a disease entity distinct from multiple sclerosis [3].

According to the current diagnostic guidelines of 2015, the presence of at least one core manifestation is sufficient for NMOSD diagnosis in anti-AQP4 seropositive patients, which are now considered classical neuromyelitis optica patients, while two manifestations including optic neuritis, myelitis or area postrema syndrome are required for seronegative patients [4]. Within the population of AQP4 negative patients, a subset has been identified expressing antibodies against myelin oligodendrocyte glycoprotein (MOG) [5]. Based on clinical, radiological and immunologic features, this condition, now called ‘MOG-immunoglobulin G (IgG)-associated encephalomyelitis’, is considered a separate disease entity, distinct from AQP4-positive neuromyelitis optica as well as from multiple sclerosis [6,7].

## 2. The Role of B Cells and Antibodies in NMOSD

The integral transmembrane protein AQP4 was discovered in 1984 and is capable of conducting water through the cell membrane. It is the most abundant type of water channel in the CNS, located specifically at the astrocytic foot processes, which form the blood-brain barrier (BBB), but also at ependyma and glia limitans [8]. Outside the CNS, it can be found in lower concentrations on the epithelial cells of many organs like kidney and stomach, but AQP4 tetramers aggregate different than within the CNS [9]. According to the current disease concept of seropositive NMOSD, autoantibodies directly targeting AQP4 penetrate the BBB or derive from peripheral plasmablasts within the CNS: Pathogenic binding of AQP4-IgG to AQP4 causes complement-dependent cytotoxicity and subsequent chemotaxis of eosinophils, neutrophils, macrophages and lymphocytes. This leads to a direct destruction of astrocytes with a secondary, irreversible demise of oligodendrocytes, axons and neurons [10]. In contrast to multiple sclerosis, cortical demyelination is only seen after years of ongoing disease activity [11]. Serum AQP4 autoantibody titers were shown to correlate with the size of spinal cord lesions, clinical disease activity and therapeutic response, as they decrease after immunotherapy and remain low during remissions [12].

Since antibody-producing cells like plasmablasts and plasma cells are derived from B lymphocytes, the role of this very lymphocyte compartment is crucial for the pathophysiology of NMOSD [13]. This goes along with the finding that in affected patients, the number of antibody-producing plasmablasts is strongly elevated in the peripheral blood, peaking at relapses [14]. It is worth noting that, in contrast to multiple sclerosis, plasmablasts can barely be detected in the cerebrospinal fluid (CSF) of NMOSD patients [15,16]. Accordingly, NMOSD is now considered a humoral autoimmune disorder, where anti-AQP4 antibodies are mainly produced in the periphery [10,17]. In line, oligoclonal bands are rarely seen in NMOSD patients, often disappearing during disease development [8]. An inflammation-induced opening of the BBB is supposed to be a prerequisite for the entry of peripherally secreted autoantibodies into the CNS. This is supported by the observation that a subset of NMOSD patients shows signs of viral infections just prior to clinical relapses [18]. Astonishingly, the permeability of the BBB has also been reported to be increased by IgG antibodies themselves [19]. Upon binding to their target on the astrocytic endfeet, AQP4 antibodies trigger the complement cascade, leading to the formation of a membrane attack complex with subsequent astrocytic edema, dysfunction, destruction and secondary neuronal injury [20]. It must be noted that T cells are also thought to be required for the disruption of the BBB and the orchestration of the immune attack in the CNS [21].

Besides antibody-mediated effects, B lymphocytes also inherit two potent, cellular functions contributing to disease activity in NMOSD: Firstly, activated B cells are capable of secreting pro-inflammatory cytokines like interleukin-(IL-)6 and tumor necrosis factor, further fostering inflammatory processes, especially by generating T helper 17 cells [22,23]. Interestingly, IL-6 in turn also supports B cell activation and growth [23]. NMOSD patients indeed display elevated IL-6 in the CSF along with an elevated number of circulating T helper 17 and IL-17-producing cytotoxic T cells [21,24]. Secondly, B lymphocytes act as antigen-presenting cells by specifically binding antigens on their B cell receptor, before internalizing, processing and presenting them to T helper cells via major histocompatibility factor II molecules on their surface [25]. It was shown that the antigen-presenting function of B cells is required for the induction of experimental CNS autoimmunity independent of their humoral involvement [26].

In conclusion, B cells uniquely contribute to disease progression in NMOSD by the provision of autoantibodies, the secretion of cytokines and the presentation of antigen. This placed in focus, we here review the interactions of rapidly evolving current and future NMOSD pharmaceuticals with B lymphocytes and autoantibodies.

## 3. Current and Evolving Therapeutic Strategies in NMOSD

Until very recently, off-label therapies were the only treatment options in NMOSD with unknown benefits and side effects. The choice of an appropriate therapy in order to prevent clinical relapses is essential to avoid irreversible disability progression, especially as recovery from clinical relapses is often incomplete in NMOSD [27]. Unfortunately, several effective drugs in the treatment of multiple sclerosis are not effective or even harmful when deliberately or unintendingly prescribed to NMOSD patients (alemtuzumab, dimethyl fumarate, fingolimod, glatiramer acetate, interferon-β and natalizumab) [28,29]. This emphasizes the importance of professional and correct diagnostic procedures before the initiation of therapeutic interventions in neuroinflammatory diseases.

The following section reviews current and potential future pharmaceutical treatment options in NMOSD with a focus on B cells, which, as shown above, are thought to be major contributors to NMOSD pathophysiology. While general immunosuppression used to be the standard therapy for many years, recently evolving treatment options include the targeted depletion of B cells, IL-6, complement antagonism and many more. Table 1 summarizes the currently available data on clinical trials in NMOSD, Figure 1 provides a graphical overview of the most important treatment strategies.

### 3.1. Immunosuppression

Although the involvement of specific AQP4 antibodies in NMOSD was uncovered only 16 years ago, an autoimmune pathogenesis of NMOSD was assumed before and several immunosuppressive agents have been tested within the last decades [30]. However, due to inconsistent diagnostic criteria, small case and open label studies, no formal approval for NMOSD treatment was granted to any of those substances.

#### 3.1.1. Glucocorticoids

In NMOSD, acute relapses are usually treated for three to five days with a pulsed intravenous methylprednisolone therapy. In severe and steroid refractory cases, plasma exchange alone or in combination with glucocorticoids is often applied as escalation therapy [31,32]. Sometimes, a prolonged oral methylprednisolone therapy is used to ‘bridge’ patients until preventive immunomodulatory or immunosuppressive treatment with one of the below mentioned agents is initiated and effective.

Glucocorticoids cause an acute lymphopenia due to a redistribution of lymphocytes to lymph nodes, spleen and bone marrow [33,34]. While no immediate change in the number of circulating B cells was observed, they were found to be slightly reduced upon prolonged glucocorticoid administration [35,36]. Interestingly, a recent study revealed a decrease of naïve B cells and an increase of memory B cells after six months of glucocorticoid treatment in IgG4-related disease [37]. However, the number of circulating plasmablasts decreased upon treatment, possibly explaining beneficial effects in NMOSD [38]. Accumulating data demonstrate that glucocorticoids down-regulate interleukin-6 serum levels [39]. It has been reported that IgG and IgA levels decrease by up to 20 percent over the first few weeks of short-term moderate-to-high dose glucocorticoid treatment, then return to normal levels over weeks to months [40,41]. In immune thrombocytopenic purpura, glucocorticoid treatment significantly reduced B cell activating factor (BAFF) serum levels after long-term treatment [42].

#### 3.1.2. Azathioprine

Although not officially approved, this widely-used purine synthesis inhibitor is traditionally considered a first-line preventive treatment of NMOSD due to substantially reduced relapse rates and disability after long-term treatment [43,44]. However, poor tolerability and a delayed mode of action cause discontinuation rates of up to 50% after 18 months of treatment [45].

Its active metabolites methylthioinosine monophosphate and thiodeoxyguanosine triphosphate inhibit purine synthesis in metabolic active cells, causing a dose-dependent decrease of total lymphocyte and B cell counts for several weeks to months [46,47,48]. While plasmablast counts seem to remain unchanged, transitional and memory B cell counts as well as IL-6 and immunoglobulin serum levels were observed to significantly drop upon azathioprine treatment [49,50,51,52,53]. The opposite is the case for serum BAFF levels, which were reported to increase in patients receiving azathioprine [54].

#### 3.1.3. Cyclophosphamide

In a limited number of patients, this alkylate showed beneficial effects in case reports and a small retrospective study of four patients with a median improvement in the expanded disability status scale (EDSS) score from 8.0 to 5.75 [55,56]. In a case report about a patient with systemic lupus erythematosus-associated NMOSD, cyclophosphamide successfully halted relapses [57]. However, other studies report adverse effects in 80% of treated patients with no significant improvement of the EDSS score [58]. This explains why cyclophosphamide is only an alternative choice for very difficult cases [59].

Cyclophosphamide is immunosuppressive but not myeloablative as the active metabolite phosphoramide mustard forms deoxyribonucleic acid crosslinks in cells with low levels of the differentially expressed enzyme aldehyde dehydrogenase [60,61]. Along with strongly decreased lymphocyte and B cell counts, an inhibition of B cell activation, proliferation and differentiation was observed after intravenous cyclophosphamide administration. After longer treatment periods, cyclophosphamide was also found to reduce memory B cell and plasmablast counts [62,63,64]. Hypogammaglobulinemia in systemic lupus erythematosus (SLE) and increased BAFF serum levels in cancer chemoimmunotherapy are thought to be a consequence of cyclophosphamide treatment [65].

#### 3.1.4. Mitoxantrone

According to the Neuromyelitis Optica Study Group, this type II topoisomerase inhibitor may be considered as second-line treatment [66]. Mitoxantrone improved EDSS scores in a small study with only five patients, but two patients experienced a relapse [67]. In a more recent, larger study with 20 patients and in line with another report, median annualized relapse rates declined from 2.8 to 0.7 and mean EDSS scores declined from 5.6 to 4.4 under mitoxantrone treatment [68,69].

By disrupting deoxyribonucleic acid synthesis and repair, this intravenous agent causes a strong decrease of lymphocyte, B cell and memory B cell counts [70,71,72,73]. While tumor necrosis factor alpha production of B cells was inhibited, the proportion of IL-10-producing regulatory B cell subsets was increased in mitoxantrone-treated patients [74]. Antibody-production was shown to be reduced and BAFF levels were increased in two other reports [74,75].

#### 3.1.5. Mycophenolate Mofetil

Although this agent requires an extended period of time to affect the immune system and is discontinued by a significant proportion of treated patients due to side effects, mycophenolate mofetil effectively reduced relapse rates and EDSS after 24 months in a prospective study with 67 patients [76]. Similar observations were made in a retrospective study with 59 included patients [77]

By inhibiting the enzyme inosine-5′-monophosphate dehydrogenase, this agent suppresses the proliferation of B and T lymphocytes, causing a reduction of lymphocyte and B cell counts [78]. In vitro, antibody formation by polyclonally activated human B lymphocytes was almost completely inhibited by its active metabolite [79]. In another study, its active metabolite inhibited T cell-induced B cell activation, B cell expansion and plasma cell differentiation in vitro. Moreover, the authors found that inhibition of B cell activation and differentiation was not due to an induction of cell death but rather due to a cell cycle arrest. However, terminally differentiated plasma cells were not susceptible to inhibition as they express reduced levels of inosine-5′-monophosphate dehydrogenase. B cells for these experiments were isolated from rheumatoid arthritis patients [80]. Similar findings in patients with systemic lupus erythematosus showed that mycophenolic acid counteracts B cell proliferation and plasmablast formation [81]. In an experimental murine model of colitis, mycophenolate mofetil promoted down-regulation of expanded B cells and production of TNF-α [82]. In SLE patients treated with mycophenolate mofetil, serum BAFF levels were increased [54].

#### 3.1.6. Methotrexate

In a small observational study including 51 patients, methotrexate dramatically decreased the frequency of relapses [69]. According to two other retrospective observational case series, the median annualized relapse rate was significantly reduced following treatment, suggesting that methotrexate is a good alternative to azathioprine [83,84]. In a retrospective assessment of treatment-recalcitrant, fulminant inflammatory CNS syndromes including NMOSD, treatment with high doses of methotrexate was observed to be safe and highly effective, producing a rapid and nearly complete cessation of disease activity [85].

Apart from the killing of proliferating cells through folate antagonism, potentially relevant modes of action of methotrexate in the treatment of autoinflammatory diseases include the inhibition of enzymes involved in purine metabolism, the inhibition of T cell activation, the selective downregulation of B cells and an increased cluster of differentiation (CD) 95 sensitivity of activated T cells [86]. Lately, circulating regulatory B cells were proposed as biomarkers for the therapeutic response to methotrexate in early rheumatoid arthritis [87]. Methotrexate has been associated with the inhibition of pro-inflammatory cytokines such as IL-1, IL-6 and tumor necrosis factor (TNF)-α [88]. In systemic lupus erythematosus, methotrexate treatment increased BAFF serum levels [54].

#### 3.1.7. Intravenous Immunoglobulins

These preparations, comprised of pooled IgG antibodies from the serum of thousands of donors, have been proven to be effective in some antibody-mediated autoimmune diseases [89]. In a study with eight patients, it was observed that the mean relapse rate and the EDSS score declined significantly upon intravenous IgG treatment in NMOSD patients [90]. Another study with only six patients found a significant reduction of relapses, while EDSS scores did not change upon therapy with intravenous IgG [91]. For acute relapses, a sole intravenous IgG infusion was not recommended as first-line therapy option, while adding high-dosed steroids to immunoglobulin therapy was superior to high-dose steroids alone in patients with high onset EDSS scores [92].

After infusion, a reduction of lymphocyte counts has been reported [93]. In patients with Guillain–Barré syndrome, B cell counts were not significantly altered by intravenous immunoglobulins [94]. Another report observed a weak decrease in blood B cell numbers accompanied with a reduced B cell proliferation [95,96]. It was shown that the frequency of plasmablasts was temporarily elevated, peaking one week after infusion, while the frequency of interleukin-6 producing B cells and the expression of CD25, CD40, CD80 and major histocompatibility complex class II on B cells was decreased [94,97,98,99]. In line, the antigen-presenting function of B cells was found to be inhibited by intravenous IgG treatment [100]. While the frequency of memory B cells remained unaltered, IL-10 producing regulatory B cells were elevated and interleukin-6 serum levels decreased [94,101,102]. Inhibition of B cell immunoglobulin production by intravenous IgG was first described in vitro [103]. In addition, decreased IgE serum levels were observed in children with asthma after intravenous IgG treatment [104]. Interestingly, intravenous immunoglobulins were shown to contain certain IgG antibodies that were able to recognize BAFF. After specific antibody binding to BAFF, steric hindrance prevented the antiapoptotic effects of BAFF on B lymphocytes [105].

### 3.2. Anti-CD20 Antibodies

#### 3.2.1. Rituximab

This widely-used monoclonal antibody binds to the CD20 surface protein on B cells causing antibody-dependent cell-mediated cytotoxicity as well as complement-dependent cytotoxicity [122]. Studies have shown profound beneficial effects on the annualized relapse rate and disability progression in NMOSD [123,124]. In a recent meta-analysis with 577 patients, the mean annualized relapse rate ratio was diminished by 1.56 and the mean EDSS score was reduced by -1.16 after rituximab therapy [125,126]. At the same time, rituximab therapy showed an overall good safety profile and an acceptable tolerability. However, one must also mention serious safety issues, including hypogammaglobinemia and prolonged neutropenia after long-term use [125].

Usually, a fast and complete depletion of circulating B cells in the blood compartment is observed upon treatment with rituximab, but the time course of B cell repopulation can vary considerably between patients. In most patients, absolute blood B cell counts are still low six to nine months after infusion [127,128]. Interestingly, it was recently shown that in the reappearing B cell pool, the frequency of immature B cells and the expression of activation markers was found to be increased [129,130]. Mature plasmablasts and plasma cells are largely preserved as they lack CD20 expression [127]. In SLE and rheumatoid arthritis patients, BAFF levels were found to be elevated during B-cell depletion and declined upon B-cell repopulation [131,132]. B cells returning after depletion secreted reduced levels of IL-6 in humans and IL-6 serum levels were shown to be unaffected six months after B cell depletion [22,133]. Further, the depletion of regulatory B cells was associated with an enhanced secretion of pro-inflammatory cytokines by CD11b^+^ antigen-presenting cells [134].

#### 3.2.2. Ocrelizumab

This humanized anti-CD20 antibody has been recently approved for the treatment of relapsing-remitting MS and binds to an epitope that overlaps the one rituximab binds [122,128]. To date, there is no published evidence regarding the treatment of NMOSD with ocrelizumab, but the same mode of action suggests a similar effectivity. As in rituximab, decreases of immunoglobulin levels were seen in ocrelizumab-treated patients [135].

B-cell-depleting therapy after ocrelizumab infusion with an interval of at least 5 months led to transiently reduced absolute lymphocyte counts, which recovered after the second cycle [126]. Effects of ocrelizumab on BAFF or IL-6 levels have not been published so far.

#### 3.2.3. Ofatumumab

This entirely human anti-CD20 antibody targets an epitope completely distinct from that of rituximab. Preceding its approval in the treatment of chronic lymphocytic leukemia, ofatumumab has been broadly investigated upon in the field of hematology and there are several ongoing studies regarding the treatment of B cell malignancies [136]. Besides an intravenous application, a subcutaneous route of administration has been tested in multiple sclerosis and rheumatoid arthritis [137,138]. In the MIRROR study, a dose-dependent B cell depletion after treatment with subcutaneous ofatumumab was observed and the efficacy of incomplete B cell depletion in multiple sclerosis was postulated [137]. Regarding NMOSD, there is a promising report about a marked reduction in relapse rate in a single patient [139].

#### 3.2.4. Ublituximab

This novel monoclonal antibody is directed against a unique epitope of the CD20 antigen. It was glycoengineered for enhanced B cell targeting through antibody-dependent cellular cytotoxicity, which allows lower doses and shorter infusion times in comparison with other anti-CD20 monoclonal antibodies [140]. In a first Phase 1 open-label, standard-of-care, single treatment arm, unblinded, single center interventional trial including NMOSD patients, experimental subjects received one infusion of 450 mg of intravenous ublituximab at the onset of an NMOSD exacerbation in addition to standard of care treatment with daily intravenous glucocorticoids at a dose of 1000 mg for five days. B cell depletion for two months was achieved in four of five patients along with improved disability status scores [108].

#### 3.2.5. BAT4406F

BAT4406F is a novel, fully humanized anti-CD20 monoclonal antibody. In an ongoing, Phase 1, open-label, dose-escalation study in NMOSD patients, the overall objective is to assess the safety, tolerability, and pharmacokinetics of BAT4406F injection in NMOSD patients (*NCT04146285*).

### 3.3. Targeting CD19

#### 3.3.1. Inebilizumab

Targeting CD19, which is expressed for longer time on differentiated B cells than CD20, this antibody leads to a broad depletion of late B cell stages including plasmablasts and early plasma cells, which are thought to produce autoantibodies in NMOSD [141]. However, it must be noted, that CD19 is not expressed on terminally differentiated plasma cells. In a recent phase 2/3 trial, inebilizumab remarkably reduced the relapse rate by 73% in both seropositive and seronegative NMOSD patients and reduced disability worsening by 15.5% along with a favorable safety and tolerability profile [115]. In June 2020, inebilizumab received its first global approval in the USA for the treatment of NMOSD in adult seropositive patients [142].

#### 3.3.2. Tandem Chimeric Antigen Receptors (CAR) T Cells Targeting CD19 and CD20

Combining both antigen-binding and T-cell activating functions into a single receptor, CAR are receptor proteins that have been engineered to give T cells the new ability to target a specific protein. Sustained B cell depletion by CD19/20-targeted CAR T cells is a highly effective treatment for murine lupus erythematosus and human acute B cell lymphoblastic leukemia [143,144]. Following this rationale, a promising phase 1 study tested safety and efficacy of anti-CD19/20 CAR T cells in the treatment of patients with AQP4-IgG seropositive NMOSD but was withdrawn due to insufficient patient recruitment (*NCT03605238*).

### 3.4. Brutons Tyrosine Kinase Inhibition

#### 3.4.1. SHR1459

Bruton’s tyrosine kinase plays a crucial role in B cell development by transmitting intracellular signals from the pre-B cell receptor [145]. There are approvals for ibrutinib in chronic lymphocytic lymphoma and acalabrutinib as well as zanubrutinib in mantle cell lymphoma [146,147,148]. The inhibition of Bruton’s tyrosine kinase has lately shown benefits in other autoimmune diseases like multiple sclerosis and systemic lupus erythematosus [149]. Following this rationale, a very recently initiated study (NCT04670770) will investigate the effect of the Bruton’s tyrosine inhibitor SHR1459 in NMOSD patients for the first time.

### 3.5. IL-6 Receptor Antagonism

#### 3.5.1. Tocilizumab

This monoclonal antibody specifically targets the IL-6 receptor, which is expressed on most immune cell subsets. IL-6 is produced by a large number of cells, including B and T cells, monocytes and fibroblasts and is considered a pro-inflammatory cytokine [150]. It has been found to be elevated in the CSF and serum of NMOSD patients and is thought to induce T cell activation, generation of autoreactive T helper 17 cells and AQP4-antibody secretion by plasmablasts [151,152]. Tocilizumab was shown to decrease the annualized relapse rate from 2.9 to 0.4 and significantly reduced EDSS, neuropathic pain, and general fatigue [153]. In a similar report, tocilizumab significantly decreased the annualized relapse rate from 4.0 to 0.4 and the median EDSS from 7.3 to 5.5 [154]. A larger, more recent trial with 118 included patients found that 92% of NMOSD patients treated with tocilizumab were relapse-free after 48 weeks compared to 69% in the azathioprine group with similar findings regarding disability progression [155]. A recent randomized phase 2 study found that tocilizumab significantly reduced the risk of a subsequent NMOSD relapse and was superior compared to with a treatment with azathioprine [117].

Treatment with tocilizumab had neither an impact on total lymphocyte count and absolute B cell, transitional B cell and plasmablast numbers, nor did it alter BAFF levels of treated patients [156,157]. However, tocilizumab was shown to reduce memory B cell subsets as well as IgA and IgG serum levels in rheumatoid arthritis patients [158]. Interestingly, it was found that tocilizumab treatment does not change interleukin-6 serum levels in rheumatoid arthritis patients [159].

#### 3.5.2. Satralizumab

Using antibody-recycling technology, this IL-6 receptor-antagonizing antibody was designed to have improved pharmacokinetics (longer half lifetime in vivo) compared to tocilizumab [160]. In two large phase 3 studies, satralizumab significantly reduced the risk of protocol-defined relapses but did not differ from placebo in its effect on pain or fatigue. Also, satralizumab was significantly less effective in seronegative patients, which requires further analysis [118,119]. In June 2020, based on these positive results, subcutaneous satralizumab received its first global approval in Canada for the treatment of NMOSD in adults and children aged ≥ 12 years who are AQP4-IgG seropositive. Satralizumab was subsequently approved in Japan and Switzerland. Satralizumab is under regulatory review in the European Union and United States of America, and is undergoing clinical development in several countries worldwide [161]. Its effects on B cells may be similar to those of tocilizumab, but still need to be examined.

### 3.6. Complement Factor Antagonism

#### 3.6.1. Eculizumab

Preventing the cleavage of complement factor 5 (C5) into its subunits, this monoclonal antibody is thought to block the AQP4-antibody-triggered complement cascade in NMOSD. Thus, inflammation, the formation of the membrane attack complex along with subsequent astrocyte destruction and neuronal malfunction are suppressed by eculizumab [20,162]. Eculizumab has been approved for the treatment of myasthenia gravis, paroxysmal nocturnal haemoglobinuria and atypical haemolytic uraemic syndrome for longer time [163]. Assuming beneficial effects in NMOSD, a phase 3 trial including 143 AQP4-antibody positive NMOSD patients, finally found that eculizumab significantly suppressed the adjudicated annualized relapse rate, while there were no benefits on disability progression during the short period of the trial [111]. These favorable outcomes of made eculizumab the first agent to be specifically approved for the treatment of seropositive NMOSD patients in the EU, USA, Canada and Japan in 2019 [164]. Despite these positive results, the drug requires a 35-min intravenous infusion every 2 weeks and C5 level activity begins to rise fairly rapidly two weeks after the last dose of eculizumab has been received [165]. Eculizumab decreased serum IL-6 levels in patients with paroxysmal nocturnal hemoglobinuria and increased B cell counts [166,167]. Effects on the B cell compartment in NMOSD patients are not yet assessed in detail and may strongly differ from those in paroxysmal nocturnal hemoglobinuria due to a completely different pathophysiology.

#### 3.6.2. Ravulizumab

As a successor of eculizumab, this monoclonal antibody is designed for longer therapy intervals of up to eight weeks as its structure differs in four amino acids, leading to a four-fold longer half-life compared to eculizumab [168]. It was already approved for the treatment of paroxysmal nocturnal haemoglobinuria in 2018 and is now also considered a therapeutic option for NMOSD [169]. There is an ongoing phase 3 study to evaluate the efficacy and safety of ravulizumab for the treatment of adult participants with NMOSD (*NCT04201262*). To date, the effect of C5-inhibitors on peripheral B cells remains unclear.

### 3.7. Targeting Pathological AQP4 Antibodies

#### 3.7.1. Aquaporumab

This antibody is designed to compete with AQP4-antibodies for AQP4 binding. Because of a mutated Fc portion, it does not activate complement- and cell-dependent cytotoxicity [170]. In a preclinical study, aquaporumab was already proven to efficiently compete with pathological AQP4-antibodies [171]. Very recently, high affinity aquaporumab, which was generated by affinity maturation using saturation mutagenesis, was shown to block cellular injury caused by NMOSD patient sera in AQP4-expressing cell cultures. Further development of aquaporumab for NMOSD therapy will require characterization and optimization of pharmacokinetics and other pharmacological properties, as well as evaluation of potential toxicity following long-term administration [172].

#### 3.7.2. HBM9161

This human monoclonal antibody targets the neonatal fragment crystallizable (Fc) receptor and blocks the IgG-Fc binding site. Thereby, it is thought to accelerate the degradation of IgG and significantly reduce the total serum IgG levels (including pathological anti-AQP4 IgG in NMOSD). There is a recruiting Phase 1 study investigating safety, tolerability, pharmacodynamics, and efficacy of HBM 9161 in patients with an attack of NMOSD (*NCT04227470*).

### 3.8. Other Therapeutic Strategies

#### 3.8.1. Bevacizumab

This well-established anti-vascular endothelial growth factor-antibody was tested in NMOSD patients with the intention to maintain the integrity of the BBB and thereby to block pathogenetic autoantibodies from entering the CNS. In a first small proof-of-concept trial with 10 patients, no patients required escalation to plasma exchange after high-dose corticosteroids and intravenous bevacizumab while an excellent safety profile was observed suggesting that bevacizumab might be considered for future studies as an add-on to currently available therapies [107]. It must be noted, that another investigation reported the emergence of a tumefactive demyelination during treatment with intravitreal bevacizumab [173].

#### 3.8.2. Bortezomib

By binding the catalytic site of the 26S subunit, this agent selectively inhibits proteasome activity and thereby leads to an enhanced apoptosis rate of rapidly proliferating cells, including plasma cells, possibly by preventing degradation of pro-apoptotic factors. Therefore, Bortezomib is approved for the treatment of multiple myeloma and mantle cell lymphoma [114]. Stimulation of murine astroglia with AQP4-IgG resulted in the release of pro-granulocytic chemokines and could be inhibited by bortezomib [174]. In one sole human study, four of the five patients were relapse-free during the 1-year follow-up after bortezomib treatment initiation. Remission was closely associated with the reduction of autoimmune activity reflected by a decrease in serum AQP4 titers, peripheral plasma cell counts and precursor B cells with proteasome inhibition. For patients with highly recurrent NMOSD who are unresponsive to conventional immunosuppressive treatments, bortezomib may serve as a salvage therapy. However, additional studies are required to validate its benefits [175].

#### 3.8.3. Cetirizine

The beneficial effect of this commonly used selective inhibitor of the histamine receptor 1 in a murine NMOSD model is thought to be related to a reduction of eosinophil infiltration into the CNS and lesion formation [176]. The fact that eosinophil infiltration is a prominent feature of NMOSD lesions and eosinophils were shown to be elevated in the CSF of NMOSD patients corroborates this hypothesis [177]. Astonishingly, cetirizine administration given as add-on therapy to standard treatment of NMOSD lead to a fourfold reduction of the annualized relapse rate after one year [112]. However, it must be noted that this study was performed with only 18 patients. Larger trials will be needed to confirm these results.

#### 3.8.4. Sivelestat

As an inhibitor of neutrophil elastase, this agent reduced lesion formation in animal models of NMOSD [178]. Dominant presence of neutrophils in inflammatory infiltrates in NMOSD and elevated neutrophil counts in the CSF may explain these findings [179,180].

#### 3.8.5. Telitacicept

B-lymphocyte stimulator/B cell activating factor (BLyS, BAFF) and A proliferation-inducing ligand (APRIL) are cytokines of the TNF-superfamily, which are capable of inducing proliferation and differentiation of B cells by activating classical and noncanonical NF-κB signaling pathways. Both cytokines can bind BAFF-receptor (BAFFR), as well as B cell maturation antigen (BCMA), as well as transmembrane activator and calcium-modulator and cyclophilin-ligand activator (TACI) [181]. These receptors are mainly expressed on maturated B cells. This explains why Telitaticept, a novel recombinant TACI-Fc fusion protein, can target and neutralize both BAFF and APRIL, which were found to be elevated in the sera of NMOSD patients when compared to MS patients [182]. Based on the promising results of a pivotal phase 2b clinical trial (*NCT02885610*), which included 249 adults with moderate-to-severe SLE, the FDA has granted fast-track approval to telitacicept for the treatment of patients with SLE. Safety assessments also showed a good tolerability profile for telitacicept in this population [183]. In a still ongoing phase III, randomized, placebo-controlled study, telitacicept is tested in seropositive NMOSD patients without recent immunosuppressive treatment (*NCT03330418*).

#### 3.8.6. C1-Esterase Inhibitor/Cinryze

As discussed above, there is compelling evidence for a central role of complement factors in NMOSD pathogenesis. Patients do not lack endogenous complement factor 1 (C1)-esterase inhibitor, but increased complement activation in acute lesions justifies its testing in NMOSD [184]. In a small study with 10 patients, the administration of a C1-esterase inhibitor, which is a nano filtered natural C1-esterase inhibitor concentrate, as add-on therapy to treatment with systemic corticosteroids was well tolerated and safe in acute NMOSD relapses. A promising benefit of C1-esterase inhibitor in NMOSD patients was suggested, as a reduction of neurologic damage and an improvement of clinical outcomes were observed, but due to the lack of a control group, the significance of these findings was limited [185]. However, in vitro data with human serum and in vivo data in rats suggest that the complement inhibition activity of C1-inhibitor in serum is too low to confer clinical benefit in NMOSD [106]. Also, chronically elevated endogenous plasma levels of C1-esterase inhibitor were found in NMOSD patients when compared to MS patients or controls [186].

#### 3.8.7. Tolerogenic Dendritic Cells Loaded with Myelin Peptides

Dendritic cells are located in peripheral and lymphoid tissues and are essential for homeostasis of T cell-dependent immune responses by promoting both pro- and anti-inflammatory activities in response to different stimuli. Within these cells, the so-called tolerogenic dendritic cells represent a heterogenous pool of cells with immunosuppressive effects regarding the expression of costimulatory molecules, the production of anti-inflammatory cytokines such as IL-10 and a reduced capacity to stimulate a T-cell response [109]. For the generation of human tolerogenic dendritic cells, isolated CD14^+^ peripheral monocytes are stimulated with granulocyte-macrophage colony-stimulating factor. After harvesting untouched, suspended cells, additional treatment with lipopolysaccharides, tumor necrosis factor-alpha or interferon-gamma generates dendritic cells. Thereafter, stimulation with cytokines (e.g. interleukin-10), organic molecules (e.g. vitamin D3), chemicals, drugs (e.g. dexamethasone) and other factors can turn them into tolerogenic dendritic cells without the existence of a standardized protocol [187]. The use of tolerogenic dendritic cells in several phase I clinical trials (type I diabetes, rheumatoid arthritis and Crohn’s disease) was safe with promising clinical and immunomodulatory results [188]. A first in human study to assess the tolerability and safety profile of a treatment with dendritic cell in patients with MS or NMOSD including 20 patients has just been completed (*NCT02283671*).

#### 3.8.8. Hematopoietic Stem Cell Transplantation

Autologous hematopoietic stem cell transplantation is a treatment option occasionally used in severe cases of NMOSD. Currently available data suggest that inflammatory activity is reduced in the short term, but a clear majority of the patients will relapse within 5 years [189]. A recent study with 11 NMOSD patients found prolonged drug-free remission with AQP4-IgG seroconversion to negative following nonmyeloablative autologous hematopoietic stem cell transplantation, warranting further investigation [110]. A more recent meta-analysis investigating the effect of autologous hematopoietic stem cell transplantation on NMOSD and including a total of 31 patients computed progression-free survival in 76% and transplant-related mortality in 0% of treated subjects [190]. Low- and intermediate-intensity regimens and RRMS patients showed optimal benefit from this therapy. Still, comparatively severe side effects and periprocedural complications further emphasize the need for less dangerous treatment options.

Allogenic hematopoietic stem cell transplantation with emerging therapeutic effects such as an induction of tolerance to auto-antigens and graft versus autoimmunity effects may point toward future treatment perspectives in this field [191].

#### 3.8.9. Mesenchymal Stromal Cells

Derived from the human umbilical cord, mesenchymal stromal cells are able to produce many different immunomodulatory molecules. In doing so, they were shown to suppress B cell proliferation through an arrest in the G0/G1 phase of the cell cycle suggesting beneficial effects in B cell-mediated autoimmune disorders [192]. In a first small study with 5 NMOSD patients, clinical disease parameters improved and relapse frequencies were reduced after initial intravenous infusion of 4 × 10^7^ human umbilical cord-derived mesenchymal stromal cells followed by 2 × 10^7^ cells both intravenously and intrathecally on days 7, 14 and 21 [193]. Here, B cells were inhibited while T cells increased after treatment, indicating an immune-related mechanism. In a case report, an NMOSD patient consented for local application of autologous mesenchymal stromal cells on pressure ulcers surprisingly showed an improvement of disability for 6 years and was free of relapses [194].

#### 3.8.10. Dalfampridine

By magnifying the axonal conductance, this potassium channel inhibitor is used in MS patients to improve cognition, fatigue, and dexterity [195]. In a randomized, double-blind, placebo-controlled crossover study of dalfampridine in transverse myelitis patients, intervention improved walking speed and other neurological functions [113]. This suggests that NMOSD patients with gait affecting transverse myelitis might also benefit from dalfampridine treatment.

#### 3.8.11. Alpha-1 Antitrypsin

A phase 1 study evaluated whether the use of alpha1-antitrypsin, an approved medication for patients with congenital deficiency of alpha-1 antitrypsin associated with emphysema, is beneficial in acute attacks of NMO, improving patient disability and quality of life (*NCT02087813*). However, it was withdrawn without any results due to recruitment issues.

## 4. Outlook

Beyond the listed agents, which are already being investigated upon in clinical trials, several promising novel treatment strategies should be monitored in the future:

### 4.1. Calcineurin Inhibition

In 2014, a patient with neuromyelitis optica spectrum disorder combined with Sjögren’s syndrome was found to be relapse-free following tacrolimus treatment [196]. Following this case, a first study with 25 included patients revealed that tacrolimus could reduce the relapse rate by 86.2% and improve the EDSS score (4.5 vs. 2.3; *p* < 0.001) significantly [197]. In a more recent report, the combined use of the calcineurin inhibitor tacrolimus with prednisolone clearly suppressed relapses in both anti-AQP4 antibody-positive and -negative NMOSD [198].

### 4.2. Targeting Eosinophils

#### 4.2.1. Anti-IL5 Antibodies

As mentioned above, eosinophil counts were found to be elevated in the CSF of NMOSD patients and eosinophil infiltration is a main feature of intracerebral lesions [177]. In this line, the second-generation antihistamine ketotifen greatly reduced AQP4-IgG/eosinophil-dependent cytotoxicity and NMOSD pathology in mice [176]. In experimental models of NMOSD, the inhibition of eosinophil granulocytes by anti-IL-5 antibodies or gene depletion significantly reduced lesion severity [176]. Therefore, IL-5 antibodies like mepolizumab and reslizumab may present promising future treatment options.

#### 4.2.2. Anti-IgE Antibodies

Case reports about highly elevated IgE-levels in patients with NMOSD support the concept of a crucial role of this immunoglobulin in NMOSD pathology [177]. The anti-IgE antibody omalizumab, already being approved for allergic asthma, might therefore also represent a considerable treatment option in patients with NMOSD [199].

### 4.3. Blocking of Pathogenic Autoantibodies

In addition to aquaporumab, a Fc-mutated anti-AQP4 antibody preventing endogenous pathogenic antibodies to bind to their target [172], small drug-like molecules like arbidol, tamarixetin and berbamine alkaloids were shown to compete with AQP4 binding and therefore may exhibit beneficial effects in NMOSD in future clinical trials [9]. Moreover, there is an interesting report about the enzyme endoglycosidase S, which specifically removes Fc glycans, rendering AQP4-antibodies non-pathogenic [200].

### 4.4. Targeting the BAFF/APRIL System

The crucial role of B cells and their effector functions in NMOSD also make them favorable targets for future therapies. Beyond the above-described telitacicept, several other monoclonal antibodies target the BAFF/APRIL complex: atacicept, an anti-BAFF antibody, surprisingly exacerbated MS disease activity in a phase 2 study [201], but may be beneficial in NMOSD because of a different pathophysiology. Other agents like tabalumab, belimumab and anti-BAFF-receptor ianalumab are already tested in clinical trials including systemic lupus erythematosus, pemphigus vulgaris, rheumatoid arthritis and multiple myeloma patients [202,203,204]. Due to the fact that they are also autoantibody-dependent diseases, NMOSD patients might also benefit from these novel agents.

### 4.5. Microbiota

The trigger of AQP4-antibody formation remains unknown up to the present day. Interestingly, AQP4-specific T cells also recognized the ABC transporter of *Clostridium perfringens*, which comprises both pathogenic and commensal strains in the human gut [205]. This is in line with the idea that recruitment and activation of autoantibody-producing B cells from the endogenous immune repertoire depends on the availability of the target autoantigen on commensal microbiota [206]. These observations identify a sequence of events triggering organ-specific autoimmune disease and these processes may offer novel therapeutic targets.

## 5. Conclusions

In summary, B cells and pathological autoantibodies are major targets when it comes to effective treatment strategies of NMOSD. Compared to MS, where B cells intrathecally secrete IgG, AQP4-tagreting IgG in NMOSD are thought to be mainly produced in the periphery, placing in focus plasmablast- and plasma cell-targeting therapies. Compared to CD20 depleting agents, inebilizumab can bind these antibody-producing cells, which might contribute to its astonishing success and approval within the last year. While general immunosuppressants inherit serious side effects, the era of monoclonal antibodies has paved the way towards a specific modification of NMOSD pathophysiology. Targeting CD19, IL-6 receptors and C5 complement has reformed the therapeutic arsenal in the rare condition of NMOSD, where clinical studies are hard to perform due to high costs, complicated diagnostic procedures and recruitment issues. The large bouquet of promising agents with differing modes of action in development may further improve the prognosis of patients suffering from this devastating disease in the future.

## Figures and Tables

**Figure 1 pharmaceuticals-14-00037-f001:**
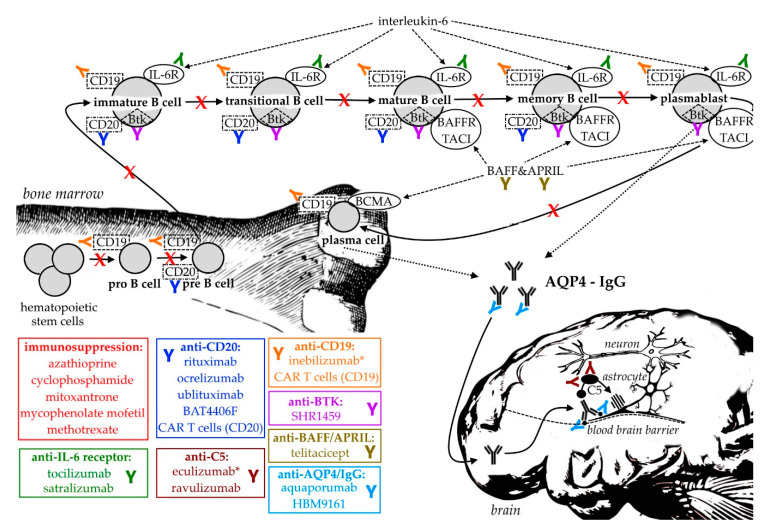
Approved (*) and experimental B cell- and antibody-targeting treatment options in neuromyelitis optica spectrum disorders. B cell development begins in the bone marrow, where hematopoietic stem cells proliferate into pro- and pre-B cells. Immature B cells, the earliest B cell subset in the peripheral blood undergo maturation into transitional, mature and memory B cells upon stimuli like interlukin-6 (IL-6), B cell activating factor (BAFF) and A proliferation-inducing ligand (APRIL). Plasmablasts and plasma cells are capable of producing immunoglobulins, including pathological anti-aquaporin-4 (AQP4) immunoglobulin G (IgG). After binding AQP4 on astrocytic foot processes at the blood-brain barrier, complement activation (including complement factor 5(C5)) causes cytotoxicity by forming membrane attack complexes with subsequent neuronal injury. BAFFR = BAFF receptor; BCMA = B cell maturation antigen; BTK = Bruton’s tyrosine kinase; CAR = chimeric antigen receptors; CD = cluster of differentiation; IL-6R = interleukin-6 receptor; TACI = transmembrane activator and calcium-modulator and cyclophilin-ligand activator.

**Table 1 pharmaceuticals-14-00037-t001:** Interventional clinical studies including neuromyelitis optica spectrum disorders (NMOSD) patients as listed on clinicaltrials.gov (*accessed December 2020*). APRIL = A proliferation-inducing ligand; BAFF = B cell activating factor; BTK = Bruton’s tyrosine kinase; C1 = complement factor 1; C5 = complement factor 5; CAR = chimeric antigen receptors; CD = cluster of differentiation; H1 = histamine receptor 1; IL-6 = interleukin-6; K^+^ = potassium; VEGF = vascular epithelial growth factor.

Clinical Trials.gov Identifier	Phase	Year Study Started	Year Approved	Intervention	Target	Participants	Status (12/2020)	Results	Section
**Phase 1**									
NCT00501748	1	2004	n/a	Rituximab	CD20	20	completed	n/a	Section 3.2.1
NCT01759602	1	2013	n/a	C1 esterase inhibitor	C1 esterase	10	completed	[106]	Section 3.8.6
NCT01777412	1	2013	n/a	Bevacizumab	VEGF	10	completed	[107]	Section 3.8.1
NCT02087813	1	2014	n/a	Alpha1-antitrypsin	n/a	0	withdrawn	n/a	Section 3.8.11
NCT02276963	1	2016	n/a	Ublituximab	CD20	6	completed	[108]	Section 3.2.4
NCT02283671	1	2015	n/a	Tolerogenic dendritic cells	n/a	10	completed	[109]	Section 3.8.7
NCT03605238	1	2018	n/a	Tandem CAR T cells	CD19/CD20	0	withdrawn	n/a	Section 3.3.2
NCT04146285	1	2019	n/a	BAT4406F	CD20	48	not yet recruiting	n/a	Section 3.2.5
NCT04227470	1	2020	n/a	HBM9161	CD20	12	recruiting	n/a	Section 3.7.2
**Phase 1/2**									
NCT00787722	1/2	2009	n/a	Autologous stem cells	n/a	13	completed	[110]	Section 3.8.8
NCT00904826	1/2	2009	2019	Eculizumab	C5	14	completed	[111]	Section 3.6.1
NCT01339455	1/2	2011	n/a	Autologous stem cells	n/a	3	terminated	n/a	Section 3.8.8
NCT01364246	1/2	2010	n/a	Mesenchymal stromal cells	n/a	20	unknown	n/a	Section 3.8.9
NCT02865018	1/2	2014	n/a	Cetirizine	H1	16	completed	[112]	Section 3.8.3
NCT03062579	1/2	2017	n/a	Tocilizumab	IL-6 receptor	10	completed	n/a	Section 3.5.1
**Phase 2**									
NCT00716066	2	2008	n/a	Autologous stem cells	n/a	40	recruiting	n/a	Section 3.8.8
NCT01845584	2	2013	n/a	Intravenous IgG	n/a	7	completed	n/a	Section 3.1.6
NCT02166346	2	2014	n/a	Dalfampridine	K^+^ channel	24	completed	[113]	Section 3.8.10
NCT02249676	2	2013	n/a	Mesenchymal stromal cells	n/a	15	completed	n/a	Section 3.8.9
NCT02893111	2	2015	n/a	Bortezomib	proteasome	5	completed	[114]	Section 3.8.2
NCT04064944	2	2019	n/a	Immunoabsorption	n/a	144	not yet recruiting	n/a	Section 3.1.1
NCT04670770	2	2020	n/a	SHR1459	BTK	10	not yet recruiting	n/a	Section 3.4.1
**Phase 2/3**									
NCT02200770	2/3	2015	2020	Inebilizumab	CD19	231	active, not recruiting	[115]	Section 3.3.1
NCT03002038	2/3	2015	n/a	Rituximab	CD20	76	completed	[116]	Section 3.2.1
NCT03350633	2/3	2017	n/a	Tocilizumab	IL-6 receptor	118	completed	[117]	Section 3.5.1
NCT03829566	2/3	2019	n/a	Autologous stem cells	n/a	0	withdrawn	n/a	Section 3.8.8
NCT04155424	2/3	2020	2019	Eculizumab	C5	15	recruiting	n/a	Section 3.6.1
**Phase 3**									
NCT00004645	3	1995	n/a	Plasma exchange	n/a	22	unknown	n/a	Section 3.1.1
NCT01892345	3	2014	2019	Eculizumab	C5	143	terminated	[111]	Section 3.6.1
NCT02003144	3	2015	2019	Eculizumab	C5	119	active, not recruiting	n/a	Section 3.6.1
NCT02028884	3	2014	n/a	Satralizumab	IL-6 receptor	83	active, not recruiting	[118]	Section 3.5.2
NCT02073279	3	2014	n/a	Satralizumab	IL-6 receptor	95	active, not recruiting	[119]	Section 3.5.2
NCT02398994	3	2015	n/a	Intravenous IgG	n/a	2	terminated	[120]	Section 3.1.6
NCT03330418	3	2017	n/a	Telitacicept	APRIL/BAFF	118	recruiting	n/a	Section 3.8.5
NCT04201262	3	2019	n/a	Ravulizumab	C5	55	recruiting	n/a	Section 3.6.2
NCT04660539	3	2021	n/a	Satralizumab	IL-6 receptor	127	not yet recruituing	n/a	Section 3.5.2
**Phase 4**									
NCT00304291	4	2001	n/a	Mitoxantrone	n/a	5	completed	[67]	Section 3.1.4
NCT02021825	4	2009	n/a	Mitoxantrone	n/a	50	unknown	n/a	Section 3.1.4
NCT02809079	4	2016	n/a	Mycophenolate mofetil	n/a	100	unknown	[121]	Section 3.1.5
NCT04256252	4	2014	n/a	Rituximab	CD20	100	terminated	n/a	Section 3.2.1

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
