# Peer review of "B Cells and Antibodies as Targets of Therapeutic Intervention in Neuromyelitis Optica Spectrum Disorders"

_pharmaceuticals, 2021, doi:10.3390/ph14010037_

Round 1
Reviewer 1 Report
No comments. Well done!
Author Response
No comments. Well done!
Thanks for this positive feedback.

Reviewer 2 Report
The review serves as a good reference for the topic chosen,.
Some suggestions for consistency and better reading.
Table 1 under intervention could have the names starting in caps and better formatting. There are some blanks which could benefit from either 0 or NA.
Suggest more consistent content on the antibodies, listing their target. Good also to mention the year started and restate the phase that the various drugs are in.
Other IL5 and IgE Omalizumab may benefit from having its own section even though they were mentioned.
Author Response
The review serves as a good reference for the topic chosen. Some suggestions for consistency and better reading. Table 1 under intervention could have the names starting in caps and better
formatting. There are some blanks which could benefit from either 0 or NA.
We appreciate this constructive feedback and inserted “n/a” when no information was available for the respective drug in the revised Table 1. In doing so, we hope to provide the reader a better overview and allow a better reading.
Suggest more consistent content on the antibodies, listing their target. Good also to mention the year started and restate the phase that the various drugs are in.
Thanks for this comment. We also think that listing the target of the therapeutic interventions helps keeping an overview of the listed agents. Thus, the revised Table 1 now contains a column which displays the target. Also, we added information about the year the study started, the phase the study is in and (if applicable) the year of approval, as suggested by another reviewer.
Other IL5 and IgE Omalizumab may benefit from having its own section even though they were mentioned.
We also think that this is a very good suggestion, thanks for this comment. By dividing the paragraph into two separate chapters, we hope to help the reader distinguish anti IL-5 and anti-IgE therapies, which may be promising agents in the future:
“4.2 Targeting eosinophils
4.2.1 Anti-IL5 antibodies
As mentioned above, eosinophil counts were found to be elevated in the CSF of NMOSD patients and eosinophil infiltration is a main feature of intracerebral lesions [180]. In this line, the second generation antihistamine ketotifen greatly reduced AQP4-IgG/eosinophil-dependent cytotoxicity and NMOSD pathology in mice [179]. In experimental models of NMOSD, the inhibition of eosinophil
granulocytes by anti-IL-5 antibodies or gene depletion significantly reduced lesion severity [179]. Therefore, IL-5 antibodies like mepolizumab and reslizumab may present promising future treatment
options.
4.2.2 Anti-IgE antibodies
Case reports about highly elevated IgE-levels in patients with NMOSD support the concept of a crucial role of this immunoglobulin in NMOSD pathology [180]. The anti-IgE antibody omalizumab, already being approved for allergic asthma, might therefore also represent a considerable treatment option in patients with NMOSD [202].” (lines 541-554)

Reviewer 3 Report
This is a welcome review on the treatment options for NMOSD based on B cells as target. The review is overall quite complete and well written.
Major points:
- The targets of the antibodies shown should be indicated in table 1 for clarity. Also drugs that are approved for NMOSD treatment as well as year of approval should be indicated (same table or a different table).
- An asset of this review is the description of the mechanisms of NMOSD pathogenesis which gives a rationale for the different drug treatments. Nonetheless, the mechanisms of the different drugs are not always specifically or clearly described. For example:
- the description of bortezomib does not state that it induces plasma cell apoptosis through proteasome inhibition. Rather the sentences at lines 427-428 are somewhat confusing.
-The BAFF-APRIL and BAFFR-TACI-BCMA pathways could be better explained, so that the possible effects of antibodies or biologicals that act on these pathway would become more clear to the reader. Also the authors do not explain that telitacicept is a TACI Fc fusion protein, which explains why it inhibits both BAFF and APRIL.
- C1-esterase inhibitor is a nanofiltered natural C1 esterase inhibitor concentrate. This should be stated.
- the authors do not explain how the tolerogenic DCs are purified or generated in vitro.
Lines 354. Write: “to have improved pharmacokinetics (longer half life in vivo) compared to tocilizumab”.
- Very little is said about the AQP4 molecule, function, is it internalized or just inhibited after autoantibody binding? Where is it expressed, only on neurons?
- There are some small imprecisions or errors:
- line 304: CD19 is expressed for longer time than CD20 on differentiating B cells but it is not expressed on terminally differentiated plasma cells.
- line 251, I don’t think that anemia is a side effect of long term rituximab treatment.
- Line 496: Mesenchymal stromal cells (now accepted term rather than stem cells) can be obtained from many tissue, not just umbilical cord (also bone marrow, adipose tissues etc). In the small study mentioned (ref 187), the authors should state the source and dose of MSCs.
-line292: monoclonal, NOT molecular
- The figures could be improved and made nicer and more precise: for example it may be possible to expand the part of the figure dealing with B cells development (perhaps incorporating BAFF-APRIL and BAFFR-TACI-BCMA as well as IL6-IL6R) and using different colours to indicate the points of inhibition of the different drugs; using only one or max 2 extended figures for this part. Then drugs acting on both B cell development and brain could be put in another figure which would include the CNS drawing.
- Sentence in lines 343-345 should be deleted as it is irrelevant to the context of the review.
Minor points:
The authors should carefully control the whole texts for typing and grammatical errors (lines 324,354,292,378,466 etc). Spacings should also be carefully checked.
Author Response
This is a welcome review on the treatment options for NMOSD based on B cells as target. The review is overall quite complete and well written. Major points: The targets of the antibodies shown should be indicated in table 1 for clarity. Also drugs that are approved for NMOSD treatment as well as year of approval should be indicated (same table or a different table).
Thanks for this constructive feedback. In the revised manuscript, Table 1 now contains both the targets of the therapeutic interventions and (if applicable) the year of approval in NMOSD. We think that this will provide the reader a more consistent and comprehensive overview of the topic. We also added a column titled “year study started” as suggested by another reviewer.
An asset of this review is the description of the mechanisms of NMOSD pathogenesis which gives a rationale for the different drug treatments. Nonetheless, the mechanisms of the different drugs are not always specifically or clearly described. For example: - the description of bortezomib does not state that it induces plasma cell apoptosis through proteasome inhibition. Rather the sentences at lines 427-428 are somewhat confusing.
We appreciate this comment. To make the effects of bortezomib more understandable, we replaced the lines 427-428 and added the following text to the paragraph:
“By binding the catalytic site of the 26S subunit, this agent selectively inhibits proteasome activity and thereby leads to an enhanced apoptosis rate of rapidly proliferating cells, including plasma cells, possibly by preventing degradation of pro-apoptotic factors. Therefore, Bortezomib is approved for the treatment of multiple myeloma and mantle cell lymphoma [114].” (lines 425-428)
-The BAFF-APRIL and BAFFR-TACI-BCMA pathways could be better explained, so that the possible effects of antibodies or biologicals that act on these pathway would become more clear to the reader. Also the authors do not explain that telitacicept is a TACI Fc fusion protein, which explains why it inhibits both BAFF and APRIL.
Thank you for this comment. We also think that a more detailed description of the BAFF/APRIL pathway may help to understand the effect of telitacicept and potential other BAFF inhibitors better. In this line, we amended the following text to the paragraph 3.8.5.:
“B-lymphocyte stimulator / B cell activating factor (BLyS / BAFF) and A proliferation-inducing ligand (APRIL) are cytokines of the TNF-superfamily, which are capable of inducing proliferation and differentiation of B cells by activating classical and noncanonical NF-κB signaling pathways. Both cytokines can bind BAFF-receptor (BAFFR), as well as B cell maturation antigen (BCMA), as well as transmembran activator and calcium-modulator and cyclophilin-ligand activator (TACI) [184]. These receptors are mainly expressed on maturated B cells. This explains why Telitaticept, a novel recombinant TACI-Fc fusion protein, can target and neutralize both BAFF and APRIL, which were found to be elevated in the sera of NMOSD patients when compared to MS patients [185].” (lines 450-457)
- C1-esterase inhibitor is a nanofiltered natural C1 esterase inhibitor concentrate. This should be stated.
This is a very helpful comment. To make the paragraph about C1-esterase inhibitors more accessible, we added the following information:
“Patients do not lack endogenous C1-esterase inhibitor, but increased complement activation in acute lesions justifies its testing in NMOSD [187]. (…) which is a nano filtered natural C1-esterase inhibitor concentrate, (…)” (lines 466-469)
- the authors do not explain how the tolerogenic DCs are purified or generated in vitro.
It is true, that information about the generation of tolerogenic dentritic cells will add value to the content of this review. Thus, we inserted the following sentences about current strategies for the in vitro generation of tolerogenic dentritic cells in paragraph 3.8.7:
“For the generation of human tolerogenic dedritic cells, isolated CD14+ peripheral monocytes are stimulated with granulocyte-macrophage colony-stimulating factor. After harvesting untouched, suspended cells, additional treatment with lipopolysaccharides, tumor necrosis factor-alpha or interferon-gamma generates dentritic cells. Thereafter, stimulation with cytokines (e.g. interleukin-10), organic molecules (e.g. vitamin D3), chemicals, drugs (e.g. dexamethasone) and other factors can turn them into tolerogenic dentritic cells without the existence of a standardized protocol [190].” (lines 483-489)
Lines 354. Write: “to have improved pharmacokinetics (longer half life in vivo) compared to tocilizumab”.
This is a good comment. We changed this sentence about satralizumab as follows:
“(…) to have improved pharmacokinetics (longer half lifetime in vivo) compared to toclizumab [163].” (lines 359-360)
Very little is said about the AQP4 molecule, function, is it internalized or just inhibited after autoantibody binding? Where is it expressed, only on neurons?
We appreciate this constructive comment. Indeed, with AQP4 being the central molecule in the current concept of the pathophysiology of seropositive NMOSD, the review will benefit from adding more background information about AQP4 to the introduction:
“The integral transmembrane protein AQP4 was discovered in 1984 and is capable of conducting water through the cell membrane. It is the most abundant type of water channel in the CNS, located specifically at the astrocytic foot processes, which form the blood–brain barrier, but also at ependyma and glia limitans [8]. Outside the CNS, it can be found in lower concentrations on the epithelial cells of many organs like kidney and stomach, but AQP4 tetramers aggregate different than within the CNS [9]. According to the current disease concept of seropositive NMOSD, autoantibodies directly targeting AQP4 penetrate the blood-brain barrier or derive from peripheral plasmablasts within the CNS: Pathogenic binding of AQP4-IgG to AQP4 causes complement-dependent cytotoxicity and subsequent chemotaxis of eosinophils, neutrophils, macrophages and lymphocytes. This leads to a direct destruction of astrocytes with a secondary, irreversible demise of oligodendrocytes, axons and neurons [10].“ (lines 52-62)
There are some small imprecisions or errors:
- line 304: CD19 is expressed for longer time than CD20 on differentiating B cells but it is not expressed on terminally differentiated plasma cells.
Thanks for this note. We changed the respective sentence to clarify this fact:
“Targeting CD19, which is expressed for longer time on differentiated B cells than CD20, (…) However, it must be noted, that CD19 is not expressed on terminally differentiated plasma cells.” (lines 312-315)
- line 251, I don’t think that anemia is a side effect of long term rituximab treatment.
This is a good point. There is indeed no reliable information about anemia in rituximab-treated patients. Thus, we deleted “anemia” in line 250.
- Line 496: Mesenchymal stromal cells (now accepted term rather than stem cells) can be obtained from many tissue, not just umbilical cord (also bone marrow, adipose tissues etc). In the small study mentioned (ref 187), the authors should state the source and dose of MSCs.
Following this comment, we replaced the term “mesenchymal stem cells” with “mesenchymal stromal cells” throughout the manuscript. Also, we stated the source and dose of the mesenchymal stromal cells to clarify the setting of the study quoted:
“In a first small study with 5 NMOSD patients, clinical disease parameters improved and relapse frequencies were reduced after initial intravenous infusion of 4x107 human umbilical cord-derived mesenchymal stromal cells followed by 2x107 cells both intravenously and intrathecally on days 7, 14 and 21 [196].” (lines 514-516(
-line292: monoclonal, NOT molecular
Sorry for this imprecision. We changed the sentence and checked the whole manuscript for similar errors:
“in comparison with other anti-CD20 monoclonal antibodies [143].” (lines 285-286)
The figures could be improved and made nicer and more precise: for example it may be possible to expand the part of the figure dealing with B cells development (perhaps incorporating BAFF-APRIL and BAFFR-TACI-BCMA as well as IL6-IL6R) and using different colours to indicate the points of inhibition of the different drugs; using only one or max 2 extended figures for this part. Then drugs acting on both B cell development and brain could be put in another figure which would include the CNS drawing.
Thanks for this idea. For reasons of simplicity and clear arrangement we tried to implement the suggestions made as possible. In doing so, we created, one unifying figure, that implements the information of all small figures in the old manuscript. It was also possible to incorporate the CNS in this figure and we hope to provide the reader a clear idea of the multiple B cell-related targets when it comes to the treatment of NMOSD. We also added BTK inhibition, as there is now also a clinical trial in phase 2, as shown in Table 1. We hope, the colors help identifying the different drugs and added the following caption to Figure 1:
“Figure 1: Approved (*) and experimental B cell- and antibody-targeting treatment options in neuromyelitis optica spectrum disorders. B cell development begins in the bone marrow, where hematopoetic stem cells proliferate into pro- and pre-B cells. Immature B cells, the earliest B cell subset in the peripheral blood undergo maturation into transitional, mature and memory B cells upon stimuli like interlukin-6 (IL-6), B cell activating factor (BAFF) and A proliferation-inducing ligand (APRIL). Plasmablasts and plasma cells are capable of producing immunoglobulins, including pathological anti-aquaporin-4 (AQP4) immunoglobulin G (IgG). After binding AQP4 on astrocytic foot processes at the blood brain barrier, complement activation (including complement factor 5(C5)) causes cytotoxicity by forming membrane attack complexes with subsequent neuronal injury. BAFFR = BAFF receptor; BCMA = B cell maturation antigen; BTK = Bruton’s tyrosine kinase; CAR = chimeric antigen receptors; CD = cluster of differentiation; IL-6R = interleukin-6 receptor; TACI = transmembran activator and calcium-modulator and cyclophilin-ligand activator.” (lines 298-309)
Sentence in lines 343-345 should be deleted as it is irrelevant to the context of the review.
Thanks for this constructive feedback. We deleted the sentence about the role of toclizumab in COVID-19, as it is indeed irrelevant to the topic.
Minor points:
The authors should carefully control the whole texts for typing and grammatical errors (lines 324,354,292,378,466 etc). Spacings should also be carefully checked.
Thanks for these hints, we apologize for the inaccuracy. We corrected the mistakes in the listed lines and checked the entire manuscript again for grammatical errors, typos and double spacing.

Round 2
Reviewer 3 Report
The authors have replied adequately to the previous comments. However there are still multiple typing errors throughout:
tocilizumab and not toclizumab in table, figure and text
line 455: transmembrane
line 484 and 489: dendritic cells and not dentritic cells
etc....
the authors should carefully check the whole manuscript and not expect reviewer or editor to correct typos.
Author Response
The authors have replied adequately to the previous comments. However there are still multiple typing errors throughout:
tocilizumab and not toclizumab in table, figure and text
line 455: transmembrane
line 484 and 489: dendritic cells and not dentritic cells
etc....
the authors should carefully check the whole manuscript and not expect reviewer or editor to correct typos.
Thanks for the positive feedback regarding the changes made in the major revision.
We apologize for the typing errors that were still in the manuscript.
“Tocilizumab” is now correctly spelled in the text, the figure and the table, the same is the case for “transmembrane” and “dendritic cells”.
In a next step, we checked the whole manuscript and mended typos in “hematopoietic”, “plasma cell”, “mitoxantrone” and “bevacizumab”. Also, inconsistent use of abbreviations was corrected.
The changes made are marked in the revised manuscript.
